# Cellular Stress Induces Nucleocytoplasmic Transport Deficits Independent of Stress Granules

**DOI:** 10.3390/biomedicines10051057

**Published:** 2022-05-03

**Authors:** Joni Vanneste, Thomas Vercruysse, Steven Boeynaems, Philip Van Damme, Dirk Daelemans, Ludo Van Den Bosch

**Affiliations:** 1KU Leuven, Department of Neurosciences, Experimental Neurology and Leuven Brain Institute (LBI), B-3000 Leuven, Belgium; joni.vanneste@kuleuven.be (J.V.); philip.vandamme@uzleuven.be (P.V.D.); 2VIB, Center for Brain and Disease Research, Laboratory of Neurobiology, B-3000 Leuven, Belgium; 3KU Leuven, Department of Microbiology, Immunology and Transplantation, Laboratory of Virology and Chemotherapy, Rega Institute for Medical Research, B-3000 Leuven, Belgium; thomas.vercruysse@kuleuven.be; 4Department of Genetics, Stanford University School of Medicine, Stanford, CA 94305, USA; sboeynae@stanford.edu; 5Department of Neurology, University Hospitals Leuven, B-3000 Leuven, Belgium

**Keywords:** neurodegeneration, amyotrophic lateral sclerosis, nuclear import, arsenite

## Abstract

Stress granules are non-membrane bound granules temporarily forming in the cytoplasm in response to stress. Proteins of the nucleocytoplasmic transport machinery were found in these stress granules and it was suggested that stress granules contribute to the nucleocytoplasmic transport defects in several neurodegenerative disorders, including amyotrophic lateral sclerosis (ALS). The aim of this study was to investigate whether there is a causal link between stress granule formation and nucleocytoplasmic transport deficits. Therefore, we uncoupled stress granule formation from cellular stress while studying nuclear import. This was carried out by preventing cells from assembling stress granules despite being subjected to cellular stress either by knocking down both G3BP1 and G3BP2 or by pharmacologically inhibiting stress granule formation. Conversely, we induced stress granules by overexpressing G3BP1 in the absence of cellular stress. In both conditions, nuclear import was not affected demonstrating that stress granule formation is not a direct cause of stress-induced nucleocytoplasmic transport deficits.

## 1. Introduction

In response to various stress conditions, eukaryotic cells decrease general translation initiation rates to allow the selective expression of proteins essential for cell survival [1]. This global translational repression—triggered by phosphorylation of the alpha subunit of the eukaryotic initiation factor 2 (eIF2α) [2,3]—causes widespread polysome disassembly. The resulting free cytoplasmic mRNA nucleates the phase separation of an array of RNA-binding proteins, such as G3BP1, into so-called stress granules [4,5,6]. These assemblies are dynamic structures and resolve upon stress recovery, concurrent with the resumption of translation.

Besides inhibiting global transcriptional regulation, cells also downregulate nucleocytoplasmic transport pathways [7]. Transport of macromolecules over the nuclear envelope passes through nuclear pore complexes (NPCs) [8] and this is facilitated by mobile nuclear transport receptors or karyopherins, also known as importins and exportins [9]. Importins bind their cargo via specific nuclear localization signals (NLSs), which are short cationic sequences enriched for lysine and arginine [9]. Various import pathways have been identified and the best characterized one is the importin-β1/importin-α_x_ pathway [9]. Importin-α subunits (7 subfamilies exist) directly bind cargoes containing a classical NLS (cNLS) and form a ternary complex with importin-β1, which mediates nuclear translocation. The most common type of nuclear export is mediated by exportin 1 (XPO1, also known as CRM1), which exports cargo proteins containing a leucine-rich nuclear export signal (NES) [9]. The directionality of transport across the nuclear membrane is driven by the RanGTP gradient [10].

Interestingly, several proteins that are part of the nucleocytoplasmic transport machinery have been found in stress granules. These include transport receptors, nucleoporins and proteins required for maintaining the Ran gradient [11,12,13,14,15,16]. This observation suggested that stress granules could contribute to stress-induced impediment of nucleocytoplasmic transport by directly sequestrating proteins of the nucleocytoplasmic transport machinery [11]. Yet, this hypothesis remains insufficiently tested.

A better understanding of the potential link between stress granules and nucleocytoplasmic transport is of particular interest to the neurodegeneration field. Stress granule proteins are found in the pathological aggregates of an expanding array of age-related neurodegenerative conditions [17], including amyotrophic lateral sclerosis (ALS) and frontotemporal dementia [18], but also spinocerebellar ataxias [19], Huntington’s disease [20] and Alzheimer’s disease [21]. Moreover, nucleocytoplasmic transport defects have also been implicated in the pathogenesis of all these conditions [22,23,24,25]. Most evidence for a convergence of these mechanisms comes from the ALS field. This disease is predominantly characterized by the cytoplasmic mislocalization and aggregation of RNA-binding proteins, such as TDP-43 [26]. As the involved proteins localize to stress granules, it has been hypothesized that these dynamic assemblies could act as stepping stones for their irreversible aggregation [18,27,28]. How TDP-43 mislocalizes to the cytoplasm is still incompletely resolved.

Hexanucleotide (GGGGCC) repeat expansions in the *C9orf72* gene are the most common genetic cause of ALS [29,30,31]. Multiple mutually non-exclusive pathogenic mechanisms have been proposed, including the formation of so-called dipeptide repeat proteins (DPRs) via uncanonical translation of the repeat RNA [32,33,34,35]. Two cationic DPRs, poly-PR and poly-GR, are potently toxic in numerous disease models. Genetic modifier screens have implicated nucleocytoplasmic transport factors in their pathogenesis [36,37,38,39], although convincing evidence which shows that the kinetics of nucleocytoplasmic transport is affected in C9-ALS is lacking [40]. Recently, we observed that nucleocytoplasmic transport was perturbed in cells that carried poly-PR-induced stress granules but not in poly-PR-expressing cells that lacked stress granules [41], suggesting that poly-PR-induced stress granules are the culprit of these nucleocytoplasmic transport defects. However, our work has shown that poly-PR needs to trigger the integrated stress response (ISR) to induce stress granule formation [42], hereby preventing us from attributing the causality to one of these two events.

To directly test whether there is a causal link between stress granules and nucleocytoplasmic transport defects, we used our recently developed nuclear import assay [41,43] to measure import rates in conditions where we uncoupled stress granule formation from ISR activation. We observed that nuclear import defects induced by arsenite occur independently from stress granule assembly, thus, arguing against the hypothesis that the interaction of nucleocytoplasmic transport factors with stress granules is required for import impediment. Our findings point at stress-induced impediment of nucleocytoplasmic transport that occurs in parallel to the formation of stress granules and disprove a causal link between them. This has important implications for our understanding of the causality of events in the neurodegenerative pathogenic cascade and has direct consequences for the search of novel therapeutic strategies to target them.

## 2. Material and Methods

### 2.1. Reporter Cell Lines

The HeLa Kyoto reporter cell lines were previously generated by our lab using CRISPR-Cas9 technology [43]. These reporter cells express a construct which exists in a green fluorescent protein carrying a NLS and NES. The reporter constructs were integrated in the AAVS1 safe harbor locus [43]. Two different reporters were used: NLS_SV40_-mNeonGreen_2x_-NES_pki2_ and NLS_c-myc_-GFP_2x_-NES_ikb2_. Cells were grown in Gibco^TM^ Dulbecco’s Modified Eagle’s Medium with high glucose (DMEM—Thermo Fisher Scientific, Brussels, Belgium; catalog number: 11965092) supplemented with 10% fetal bovine serum (FBS—GE healthcare, Machelen, Belgium), Hyclone, catalog number: SV30160.03) and Gibco^TM^ Gentamycin (20 μg/mL, Thermo Fisher Scientific, Brussels, Belgium; catalog number: 15750045).

### 2.2. Measuring Nucleocytoplasmic Transport

Nucleocytoplasmic transport was measured as described before [41,43]. Cells expressing the reporter NLS_SV40_-mNeonGreen_2x_-NES_pki2_ were incubated with the XPO1-inhibitor leptomycin B (LMB—45 nM; Invivogen, Toulouse, France; catalog code: tlrl-lep, dissolved in ethanol) for indicated time points.

To quantify nucleocytoplasmic transport, the nuclear and cytoplasmic intensity was measured either manually or automatically. For manual analysis, images were taken under a SP8 confocal microscope (64x; Leica, Wetzlar, Germany; model: SP8 MDi8) and manually analyzed with Rasband, W.S., ImageJ, U. S. National Institutes of Health, Bethesda, Maryland, USA, https://imagej.nih.gov/ij/, 1997–2018 (version 1.53h). A minimum of five images per condition were taken and about five cells per image were analyzed per experiment. The average of one image was seen as one data point. Three to four replicates were performed. For automated analysis, cells were automatically analyzed by the High Content Imager (CellInsight CX5 high-content screening platform, Thermo Fisher Scientific, Brussels, Belgium; catalog number: CX51110) and the Thermo Scientific HCS Studio 4.0 Cell Analysis Software. Three wells were analyzed per condition. The average of one well was seen as one data point. Three to four independent experiments were performed. As previously shown, the manual and automatic evaluation of nucleocytoplasmic transport resulted in similar results [41].

### 2.3. Treatment of Cells with Sodium Arsenite

Sodium arsenite (NaAsO_2_) was dissolved in Milli-Q water to obtain a stock solution of 50 mM. Hela Kyoto cells were treated with 0.5 mM, 0.1 mM or 0.05 mM of sodium arsenite for 60 min before measuring nucleocytoplasmic transport or performing immunocytochemistry. Milli-Q water was used as a vehicle control.

### 2.4. Treatment of Cells with ISRIB

Cells were preincubated with the integrated stress response inhibitor (ISRIB—2 µM, Sigma-Aldrich, Machelen, Belgium; catalog number: SML0843, dissolved in DMSO at a stock concentration of 1.11 mM) for 4 h. DMSO was used as a vehicle control.

### 2.5. Cellular Expression of mCherry, G3BP or IBB

Cells were transfected with plasmids expressing mCherry, G3BP or IBB-RFP using Lipofectamine^TM^ 3000 (Thermo Fisher Scientific, Brussels, Belgium; catalog number: L3000015) in accordance with the manufacturer’s protocol. Transfection medium was removed 4 h after transfection to limit toxicity. Cells were used for analysis after 48 h.

### 2.6. siRNA-Based Knock-Down of G3BP

Stealth siRNAs specific for human *G3BP1* RNA (Stealth Selected RNAi, siRNA ID: HSS115446) and human *G3BP2* RNA (Stealth Selected RNAi, siRNA ID: HSS114988) and negative control siRNA were purchased from Thermo Fisher (Brussels, Belgium). Targeting G3BP knock-down was based on the paper of Matsuki et al. [44]. Transfection was carried out with 100 pmol siRNAs using Lipofectamine^TM^ 2000 Transfection Reagent (Thermo Fisher Scientific, Brussels, Belgium; catalog number: 11668027), according to the manufacturer’s instructions. Transfection medium was removed 4 h after transfection to reduce toxicity. No antibiotics were added to the media during and after transfection to limit cell death. Cells were used for analysis after 48 h.

### 2.7. Immunocytochemistry

Cells were fixed with 4% paraformaldehyde (4% PFA—Polysciences Inc., Hirschberg and der Bergstrasse, Germany; catalog number: 2878-55-4) in Gibco^TM^ Dulbecco’s Phosphate-Buffered Saline (DPBS—Thermo Fisher Scientific (Brussels, Belgium), catalog number: 14190250) for 15 min and rinsed three times with DPBS. In total, 5 % normal donkey serum (NDS, Sigma, Machelen, Belgium; catalog number: D9663) in 0.1% Triton X-100 in DPBS was used for blocking at room temperature for 1 h. Primary antibodies were diluted in 2% NDS in 0.1% DPBS Triton and incubated overnight at 4 °C. The following primary antibodies were used: anti-G3BP (1/250, catalog number: ab56574), anti-Yb1 (1/500, catalog number: ab76149) and anti-importin-β1 (1/1000, catalog number: ab2811) all provided by Abcam (Cambridge, UK). The cells were subsequently washed with DPBS and incubated with appropriate secondary antibodies (1:2500; Thermo Fisher Scientific, Brussels, Belgium). Nuclei were stained with Hoechst (NucBlue Live ReadyProbes^TM^ Reagent Hoechst 33342, Thermo Fisher Scientific, Brussels, Belgium; catalog number: R37605) and images were taken under SP8 confocal microscope (64x; Leica, Wetzlar, Germany; model: SP8 MDi8) excitation lines at 405, 488, 555 and 647 nm.

### 2.8. Western Blot

For blotting of G3BP, cells were trypsinized with 0.05% Gibco^TM^ Trypsin-EDTA (Thermo Fisher Scientific, Brussels, Belgium; catalog number: 253000054) and transferred to a 15 mL tube. Cells were centrifuged at 500× *g* for 5 min, after which the supernatants were removed. The pellet was dissolved in pre-chilled RIPA buffer (50 mM Tris-HCl (pH 7.5), 150 mM NaCl, 1% NP-40, 0.5% Na-deoxycholic acid, 0.5% SDS, Thermo Fischer Scientific (Brussels, Belgium), catalog number: 89901) supplemented with cOmplete^TM^ EDTA-free Protease Inhibitor Cocktail (Sigma, Machelen Belgium; catalog number: 5056489001). Cells were transferred into a 1.5 mL Eppendorf tube and put on ice for 30 min.

Protein concentrations were measured with the Micro BCA^TM^ Protein Assay Kit (Thermo Fisher Scientific (Brussels, Belgium), catalog number: 23235). Reducing sample buffer (Thermo Fisher Scientific, Brussels, Belgium; catalog number: 39000) was added to samples containing equal amounts of protein (6 µg) and heated for 10 min at 95 °C before separation on a 4–20% Mini-PROTEAN TGX Precast Protein gel (Bio-Rad, Temse, Belgium; catalog number: 4561093) for 1 h (100 V). Proteins were transferred to a polyvinylidene difluoride (PVDF) membrane (Trans-blot Turbo^TM^ Mini PVDF transfer, Bio-Rad, Temse, Belgium; catalog number: 1704156) by Trans-Blot Turbo^TM^ Transfer system (Bio-Rad, Temse, Belgium; catalog number: 1704150). Membranes were blocked with 5% non-fat milk (Cell Signaling Technology, Danvers, MA, USA; catalog number: 9999S) in Tris Buffer Saline Solution with 0.1% tween (TBST) for 1 h at room temperature and incubated with primary antibodies in TBST overnight at 4 °C. The following antibodies were used: anti-G3BP (1/1000, Abcam, Cambridge, UK; catalog number: ab56574) and anti-GAPDH (1/10,000, Thermo Fisher Scientific, Brussels, Belgium; catalog number: AM4300). The next day the membranes were washed 3 times with TBST and incubated for 1 h with the appropriate secondary antibody conjugated with horseradish peroxidase in TBST (1/5000, Agilent Technologies, Machelen, Belgium; catalog number: P044701-2). Proteins were detected by enhanced chemiluminescence reagents (ECL substrate—Thermo Fisher Scientific, Brussels, Belgium; catalog number: 32106) and an ImageQuant LAS 4000 biomolecular Imager (GE Healthcare, Chicago, IL, USA). Luminescent signals were analyzed using ImageQuant TL software version 7.0 (GE Healthcare, Chicago, IL, USA) and normalized to the loading control.

### 2.9. Statistical Analysis

D’Agostino-Pearson omnibus normality test was used to test data for normality. Parametric tests were used on normally distributed data. Non-parametric tests were used on non-normally distributed data. Unpaired *t*-test/Mann–Whitney U-test was used to determine statistical differences between two groups. (Non)-parametric one-way ANOVA followed by Dunn’s multiple comparison/Dunnett’s multiple comparison test was used to determine statistical differences between more than two groups. Data values represent mean ± SD. When non-parametric tests were performed, data were presented as median ± IQR (when n < 25) or geometric mean ± geometric SD (when n < 25). Graphpad Prism version 8 was used to perform statistical analyses. * indicates *p* < 0.05; ** indicates *p* < 0.01; *** indicates *p* < 0.005; **** indicates *p* < 0.001.

## 3. Results

### 3.1. Arsenite Suppresses Importin-β1/α1-Mediated Nuclear Import

To investigate the role of stress granules in stress-induced transport defects, we used sodium arsenite (NaAsO_2_), which induces oxidative stress leading to eIF2α phosphorylation and stress granule formation (Figure 1) [45,46]. In line with previous results [11], both importin-β1 and importin-α1 localized into these granules, consistent with the hypothesis that nucleocytoplasmic transport could be inhibited via sequestration of essential transport factors in stress granules (Figure 1A,B). To measure stress-induced transport deficits, we focused on the conventional import pathway mediated by importin-β1/α_x_ [47]. This pathway is also responsible for the import of the ALS-associated protein TDP-43 [48]. In order to analyse importin-β1/α_x_-mediated import, we made use of two previously established reporter cell lines [41,43,49]. In short, HeLa Kyoto cells stably expressed a green fluorescent protein that carries a nuclear export signal (NES) recognized by exportin 1 (XPO1) and a nuclear localization signal (NLS) recognized by importin-β1/α_x_ (Figure 2A). The first cell line stably expressed the cargo reporter protein NLS_c-myc_-GFP_2x_-NES_ikb2_ (Figure 2A—steady state reporter). This reporter localized under steady state conditions mainly in the nucleus due to a strong NLS. Incubation of the reporter cells with arsenite resulted in an increased steady state cytoplasmic localization of the reporter (Figure 2B). This indicates that arsenite treatment decreased importin-β1/α_x_-mediated import and/or increased XPO1-mediated export [41,43].

To measure importin-β1/α_x_-mediated import separately, a second cell line was used (Figure 2A—dynamic reporter). This second cell line stably expressed a cargo reporter protein NLS_SV40_-mNeonGreen_2X_-NES_pki2_. Under control conditions, this reporter was mainly localized in the cytoplasm due to the presence of a strong NES (Figure 2A). As a consequence, inhibiting nuclear export with the XPO1-inhibitor leptomycin B (LMB) will allow the evaluation of the importin-β1/α_x_-mediated import in a dynamic way by measuring the increase in nuclear intensity over time (Figure 2C,D). When treating these reporter cells with arsenite, we observed both a significantly reduced nuclear intensity as well as a lower nuclear/cytoplasmic ratio over time (Figure 2C,D). These observations suggest that less reporter is imported over time and confirm a direct inhibitory effect on nuclear import by arsenite treatment.

Notably, we observed a strong localization of the dynamic reporter in stress granules (Figure 2E arrow 7–12). This co-localization was not caused by bleed-through of the stress granule antibody into the 488 channel because the same localization of the reporter in the stress granules was also observed without any additional staining (Appendix A). Of note, the NLS_c-myc_-GPF_2x_-NES_ikb2_ steady state reporter also showed some stress granule localization (data not shown). Why or how this reporter localized to these granules is still unclear. One possibility is that the stress granule localization of the reporter cargo protein is importin-β1/α_x_ dependent, as a reporter without an NLS showed significantly less localization into stress granules (Appendix A).

To evaluate whether the above measured decrease in nuclear import was caused by the accumulation of the reporter in the stress granules and therefore decreased the availability of the reporter, we reanalyzed the cytoplasmic intensities but excluded the reporter present in the stress granules (Figure 2F). This did not change the observed decrease in nuclear/cytoplasmic intensity (Figure 2F). Indeed, fluorescence intensity from cytoplasmic ‘free’ reporter proteins was observed in arsenite-treated cells after 30 min of LMB treatment (Figure 2E arrow 10–12). These data suggest that the observed decrease in the nuclear/cytoplasmic ratio of NLS_SV40_-mNeonGreen_2X_-NES_pki2_ reporter cells is caused by a general increase in ‘free’ cytoplasmic reporter and not by a misinterpretation of this parameter caused by the localization of the reporter at stress granules. In conclusion, these findings indicate that arsenite treatment could impede importin β1/α_x_-mediated import.

### 3.2. Inhibition of Stress Granule Formation Does Not Rescue Arsenite-Induced Transport Deficits

To investigate whether stress granules contribute to the observed nuclear transport deficits, we prevented cells from assembling stress granules despite being subjected to cellular stress. G3BP1 and G3BP2 are key nucleators of arsenite-induced stress granules [4,44] and their knock-down/knock-out has been shown to prevent stress granule assembly [44]. Indeed, siRNA-mediated knock-down of both G3BP1 and G3BP2 prevented the formation of stress granules in the majority of arsenite-treated cells, as indicated by the absence of yb1-positive granules (Figure 3A,B and Appendix A). Despite the lack of stress granules, nuclear import defects were indistinguishable between control siRNA- and G3BP siRNA-treated cells in our two reporter assays (Figure 3C,D). These observations argue against the hypothesis that stress granules directly contribute to the stress-induced decrease in nucleocytoplasmic transport. Moreover, the increased cytoplasmic localization of importin-α1 in arsenite-treated cells was also independent of stress granule formation, as it was not abolished by the knock-down of G3BP1/2 (Appendix A).

To independently confirm that stress granule formation did not affect nuclear transport, we pharmacologically inhibited stress granule formation. The integrated stress response inhibitor (ISRIB) is a compound that suppresses the integrated stress response via a mechanism downstream of eIF2α-phosphorylation [50]. As a consequence, ISRIB rescues translation and inhibits arsenite-induced stress granule formation [51]. However, ISRIB only induces these rescue effects when intracellular phosphorylation levels of eIF2α do not exceed critical threshold levels (40–70% of the maximal eIF2α phosphorylation level) [51]. Accordingly, we lowered our arsenite concentration in this experiment from 500 µM to 100 µM, as this is the highest concentration at which ISRIB blocked arsenite-induced stress granule formation [51]. We indeed confirmed that ISRIB repressed stress granule formation in the majority of cells treated with 100 µM arsenite, but not in cells treated with 500 µM arsenite (Figure 4A and Appendix A). Importantly, pre-incubation of arsenite-treated cells with ISRIB did not rescue import deficits in both reporter cell lines (Figure 4B,C). Additionally, 500 µM arsenite induced stronger transport deficits compared to 100 µM (Figure 4C and Figure 5A), while there was no difference in the size and number of stress granules between 100 µM and 500 µM of arsenite conditions (Figure 5B,C). In addition, similarly as for the 500 µM condition (Figure 1A), 100 µM arsenite stress granules were able to recruit importin-β1 (Figure 5D). In line with our G3BP siRNAs findings, these observations further indicate that stress granule formation is not critical for stress-induced nucleocytoplasmic transport deficits and does not correlate with the recruitment of importin-β1 to these stress granules.

### 3.3. Stress Granule Formation Is Not Sufficient to Induce Nuclear Import Deficits

Our results indicate that stress granules are not required to induce nuclear import impediments under oxidative stress induced by arsenite. Next, to further corroborate these findings, we tested whether stress granules per se can induce transport deficits independent of an abiotic stress stimulus by uncoupling stress granule formation from stress. As documented before [44], the overexpression of G3BP1 induced the spontaneous assembly of stress granules in the absence of stress (Figure 6A,B). Similar to the arsenite-induced stress granules (Figure 1A), importin-β1 was localized to these granules (Figure 6C). However, overexpression of G3BP induced the formation of larger stress granules compared to the ones induced by arsenite (Figure 6D). As a consequence, these stress granules were able to trap a significantly higher percentage of cytoplasmic importin-β1 (Figure 6D). On average, 12% of total importin-β1 signal localized to arsenite-induced stress granules and this more than doubled (~30%) in G3BP-overexpressing cells.

Similar to the arsenite experiments, we measured nucleocytoplasmic transport in our two reporter cell lines. We did not observe a significant effect of G3BP1 overexpression compared to mCherry on the nuclear/cytoplasmic distribution of the steady state reporter (Figure 7A). This suggests that G3BP overexpression-induced stress granules do not affect nucleocytoplasmic transport. This absence of a significant impact on the nuclear/cytoplasmic ratio was observed despite the strong localization of importin-β1 (Figure 6D) and the reporter (Figure 7B) at stress granules. This indicates that the localization of the reporter at stress granules does not influence its nucleocytoplasmic transport, which is in agreement with the observations made for arsenite-induced stress granules (Figure 2E,F).

When we measured nuclear import using the dynamic reporter cell line, we observed a significant decrease in the nuclear/cytoplasmic ratio over time (Figure 7C). However, when we assayed the nuclear intensity of the NLS_SV40_-mNeonGreen_2X_-NES_pki2_ reporter instead of the nucleocytoplasmic ratio, there was no difference between mCherry- or G3BP1-expressing cells (Figure 7D). This indicates that in both conditions the same amount of reporter is transported into the nucleus over time and argues against a decrease in nuclear import in G3BP1-expressing cells. The absence of a nuclear import defect in G3BP1-overexpressing cells indicates that the altered nucleocytoplasmic ratio stems from differences in the measured cytoplasmic intensity. Notably, we still observed residual reporter fluorescence at the site of stress granules after 30 min of LMB treatment, whereas there remained no ‘free’ reporter visible in the cytoplasm (Figure 7F arrow 10–12). This indicates that the localization of the reporter at stress granules underlies the decreased nuclear/cytoplasmic intensity measured in Figure 7C. Indeed, excluding the reporter present in stress granules when measuring cytoplasmic intensities abolished the decreased nuclear/cytoplasmic ratio (Figure 7E).

In conclusion, these findings indicate that—if large enough—stress granules can trap a significant residual pool of client proteins in the cytoplasm. Yet, given that the rate of nuclear import over time was unaltered in our experiments, there is no evidence that stress granules can directly impede nuclear import via the sequestration of nuclear import factors.

## 4. Discussion

Over the last few years, both defects in stress granule metabolism and nucleocytoplasmic transport have been suggested to be involved in the etiology of a spectrum of neurodegenerative conditions. The observation that a variety of nucleocytoplasmic transport factors are found in stress granules led to the hypothesis that these granules can directly impede transport by trapping transport factors [11,12,13,14,15,16]. Here, we confirm the recruitment of two key nuclear import factors, namely importin-β1 and importin-α1, into arsenite-induced stress granules. In addition, using two nucleocytoplasmic transport reporter cell lines we observed impeded nuclear import in these cells. However, although these findings suggest a potential role of stress granules in stress-induced impairment of nucleocytoplasmic transport, they do not directly support a causal relationship as we cannot rule out direct effects in nuclear import by ISR activation. To disentangle this, we used two independent approaches to prevent stress granule assembly under stress (G3BP1/2 knock-down and ISRIB treatment) and found that stress granules are not required for stress-induced transport deficits. This is also supported by the observation that higher arsenite concentrations induced stronger transport deficits without a significant effect on stress granule formation itself. These observations argue against a causative role for stress granules in arsenite-induced nucleocytoplasmic transport defects.

To further corroborate these findings, we investigated the ability of stress granules to inhibit nucleocytoplasmic transport in the absence of cellular stress. Overexpression of G3BP1 resulted in the formation of large stress granules that—just as their arsenic counterparts—colocalized with nuclear import factors. Using our steady state reporter line (NLS_c-myc_-GPF_2x_-NES_ikb2_), we found no differences in nuclear import. However, our dynamic reporter line (NLS_SV40_-mNeonGreen_2X_-NES_pki2_) did show a decreased nucleocytoplasmic ratio over time. While this result could point towards an actual decrease in nuclear import, we performed a number of additional control experiments to further investigate this. First, we found that there was no difference in the increase in nuclear reporter signal over time between G3BP1- and mCherry-overexpressing cells. Second, looking in more detail at the cytoplasmic component of the nucleocytoplasmic ratio, we observed that our reporter retained some residual cytoplasmic localization to stress granules, but no ‘free’ cytoplasmic reporter was left. These findings indicate that while stress granules can trap some residual cargo protein, they do not impact the rate of nuclear import of ‘free’ cargo protein.

Our data demonstrate that stress granules are not causally responsible for the decrease in transport induced by arsenite-induced stress. Moreover, stress granule formation on its own is not sufficient to decrease general nucleocytoplasmic transport. Taken together, this implies that the localization of transport factors at stress granules is not responsible for the transport defects observed in *C9orf72* poly-PR cell models, in contrast to what was previously suggested by us [41] and by others [11]. This conclusion is in line with previous findings made by Hutten et al., who observed poly-GR-induced transport deficits in the absence of stress granules [52]. In addition, this implies that stress granule formation does not underlie the cytoplasmic mislocalization of TDP-43 observed in ALS patients by a reduction in nuclear import caused by the limited availability of transport receptors.

We must note that our study has some limitations. We only made use of arsenite and G3BP1 overexpression for the induction of stress granules, while stress granule assembly can be different depending on the stressor [53]. Stress granule assembly in our model may differ from what occurs in motor neurons in ALS patients. Further research will be required to address this question. In addition, our experiments were performed in human HeLa Kyoto cells. As cell-type-specific effects cannot be excluded, it might be important to confirm these findings in additional cell types.

Our data differ from the observations made by Zhang et al., who determined that stress granule formation is directly responsible for arsenite-induced impediment of nucleocytoplasmic transport [11]. The exact reason for this discrepancy is currently unclear, as a similar approach was used to inhibit stress granule formation, namely a G3BP-based strategy as well as a compound-based strategy [11]. However, we used a G3BP knock-down strategy in comparison to the use of G3BP knock-out cells [11]. While a difference between these two strategies could potentially explain such discrepancies, the fact that we prevent stress granule assembly in more than 90% of cells, argues against a difference between knock-down and knock-out efficiency in reducing G3BP levels as an explanation for this discrepancy. For their pharmacological approach, Zhang et al. made use of the ISR inhibitor ISRIB at the same concentration (2 µM), but in cells treated with a higher arsenite concentration (500 µM) [11]. Because ISRIB did not inhibit stress granule assembly induced by 500 µM arsenite in our hands, we were not able to reproduce this experiment. A second compound used by the authors is the PERK inhibitor GSK2606414 [11], which is known to inhibit stress granule formation via inhibition of PERK-based phosphorylation of eIF2α [54]. However, we could not replicate this experiment due to the inability to inhibit stress granule formation (Appendix A). This could be due to the fact that arsenite-induced phosphorylation of eIF2α occurs independent of PERK [55].

If stress granule formation does not induce a decrease in nucleocytoplasmic transport, the question remains what the cause is of the observed stress-related transport deficits. It has been well characterized that various cellular stresses, such as heat shock and oxidative stress, can indeed downregulate nucleocytoplasmic transport pathways in various ways and independent of stress granules [56,57]. For example, nuclear retention of importin-α1 [58], the collapse of the Ran GTPase gradient [57,59,60,61,62,63] and post-translational modifications and/or redistribution of nucleoporins have been observed [57,64,65,66,67]. We observed a relocalization of importin-α1 in arsenite-treated cells. Despite the localization of importin-α1 in stress granules, its increased cytoplasmic localization was independent of stress granule formation, as it was not abolished by the knock-down of G3BP1/2 and it was not induced by G3BP1 overexpression (Appendix A). This stress granule-independent mislocalization of importin-α1 could potentially explain the observed transport deficits induced by arsenite treatment, but it is likely that additional mechanisms are at play.

Our findings argue against a direct role for stress granules in general transport defects, leaving the question unanswered of what the functional relevance of the localization of importins to stress granules is. Accumulating evidence suggests that stress granules protect aggregation-prone proteins against irreversible protein aggregation under conditions of stress [68,69]. While the high concentration of RNA within stress granules may help to chaperone these proteins [69,70], other reports highlighted the importance of chaperones in maintaining stress granule dynamics [71]. Interestingly, nuclear import factors were recently found to moonlight as protein chaperones [16,72,73,74]. Moreover, it is exactly the NLS-bearing stress granule proteins that mislocalize and aggregate in ALS which are substrates to this chaperone activity. This suggests that besides acting as a targeting sequence to the nucleus, NLSs may as well link aggregation-prone proteins to their transport receptors in cytoplasmic stress granules. In line with this, we observed that the stress granule localization of our reporter was dependent on the presence of its NLS. Of note, all these data suggest that the localization of nuclear transport factors to stress granules could be beneficial (i.e., promoting stress granule dynamics), rather than detrimental (i.e., driving transport defects).

In conclusion, we provide evidence against a causal role of stress granules in driving the stress-induced general impediment of nucleocytoplasmic transport. Instead, we hypothesize that stress-granule-independent mechanisms, including but not limited to the cellular redistribution of importin-α1, are at play. Our data strongly suggest that the mechanistic connection proposed between stress granules and nucleocytoplasmic transport defects should be reevaluated. Lastly, these findings also have an important impact on the development of therapeutic strategies, since targeting stress granule dynamics might not be sufficient to counteract nucleocytoplasmic transport defects.

## Figures and Tables

**Figure 1 biomedicines-10-01057-f001:**
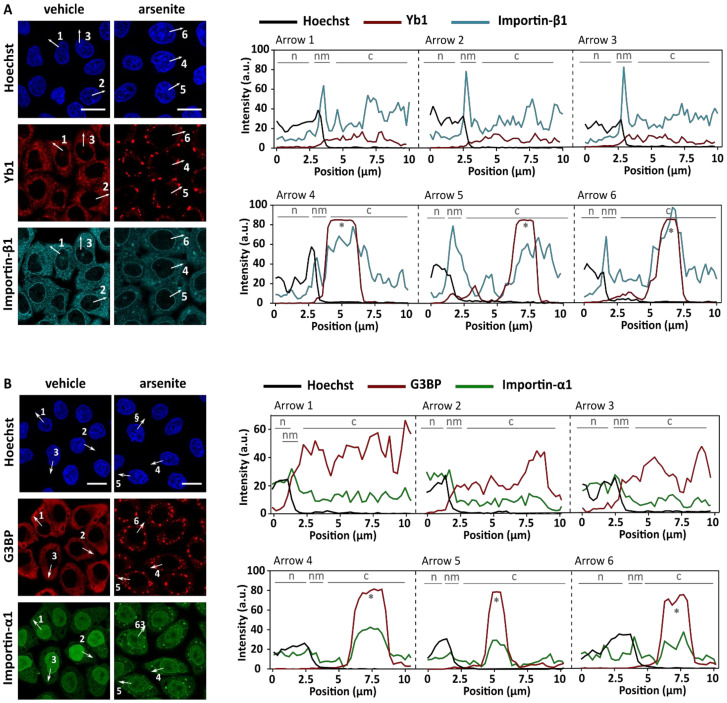
Importin-β1 and importin-α1 localize at arsenite-induced stress granules. (**A**). HeLa Kyoto cells were treated with 500 µM sodium arsenite (NaAsO_2_) for 60 min before fixation with 4% PFA. Cells were stained with an anti-Y-box binding protein (yb1) antibody (=stress granule marker) and an anti-importin-β1 antibody. Hoechst was used to visualize the nucleus. Arsenite treatment resulted in the formation of stress granules in which importin-β1 localizes, which is confirmed by the intensity plots shown on the right. N = nucleus; nm = nuclear membrane; c = cytoplasm; * = stress granule. Scale bar = 20 µm. (**B**). HeLa Kyoto cells endogenously expressing importin-α1-mNeonGreen were treated for 60 min with 500 µM arsenite and stained with an anti-G3BP antibody (stress granule marker) after fixation. Importin-α1 localized at stress granules, which is confirmed by the intensity plots shown on the right. n = nucleus; nm = nuclear membrane; c = cytoplasm; * = stress granule. Scale bar = 20 µm.

**Figure 2 biomedicines-10-01057-f002:**
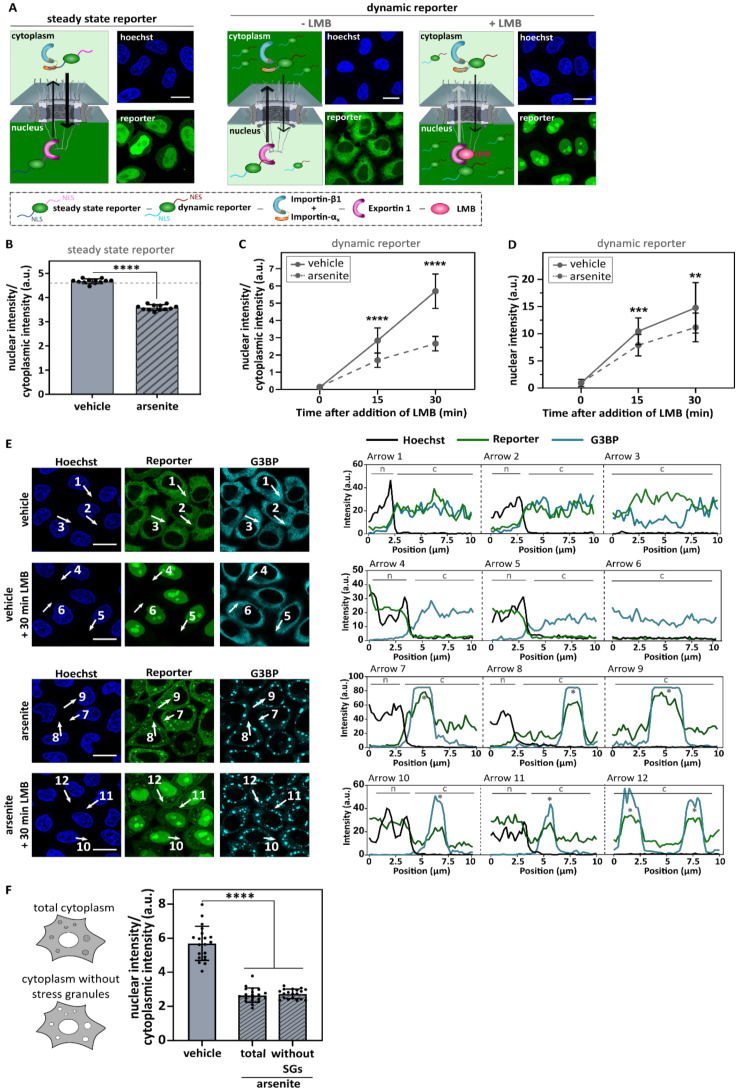
Sodium arsenite suppresses importin-β1/α1-mediated import. (**A**). HeLa Kyoto cells stably express the reporter construct NLS_c-myc_-GFP_2X_-NES_ikb2_ (left) or the reporter construct NLS_SV40_-mNeonGreen_2X_-NES_pki2_ (right), of which import is mediated by importin-β1/α_1_ and export is mediated by XPO1 [41,43]. Two copies of the fluorophore in tandem allow us to primarily focus on active transport by limiting passive transport [41,43]. Left: the ‘steady state reporter’ localizes under control conditions mainly into the nucleus, due to a stronger import. This allows the analysis of the impact of arsenite on import via measuring the nuclear/cytoplasmic ratio in steady state. Right: in control conditions the ‘dynamic reporter’ is mainly localized to the cytoplasm due to a stronger nuclear export. Consequently, inhibition of XPO1-mediated import via leptomycin B (LMB), induces a shift towards the nucleus, which allows the measurement of nuclear import over time. Scale bar = 20 µm. (**B**). Nuclear/cytoplasmic ratio was automatically measured (high-content analyzer) in HeLa Kyoto cells expressing the reporter NLS_c-myc_-GFP_2X_-NES_ikb2_ after treatment with 500 µM arsenite for 60 min. Arsenite treatment significantly disturbed nucleocytoplasmic transport. Dots represent means of one well with 200–1000 cells per well; n = 12 from four experiments with each three wells. Mann–Whitney U test. Geometric means ± geometric SD. **** indicates *p* < 0.001. (**C**). Nuclear import was measured using ImageJ in HeLa Kyoto cells expressing the reporter NLS_SV40_-mNeonGreen_2x_-NES_pki_ after treatment with 500 µM arsenite for 60 min. Arsenite treatment significantly decreased importin-β1-mediated import. Dots represent means of one image with five cells per image; n = 20 from four experiments with each five images. Unpaired *t*-test. Means ± SD. **** indicates *p* < 0.001. (**D**). Nuclear intensities of HeLa Kyoto cells expressing the reporter NLS_SV40_-mNeonGreen_2x_-NES_pki_ treated with arsenite shown in panel (**C**). Unpaired *t*-test. Means ± SD. ** indicates *p* < 0.01; *** indicates *p* < 0.005. (**E**). Representative images and corresponding intensity plots of NLS_SV40_-mNeonGreen_2x_-NES_pki_ reporter cells treated with indicated conditions. Intensity plots indicate that the reporter colocalizes with G3BP at arsenite-induced stress granules (arrow 7–12). n = nucleus; nm = nuclear membrane; c = cytoplasm; * = stress granule. Scale bar = 20 µm. (**F**). Data obtained in panel (**C**) after 30 min treatment with LMB were reanalyzed by excluding the stress granules when measuring the cytoplasmic intensity. Significant transport deficits were observed when stress granules were excluded from the analysis. One-way ANOVA followed by Dunnett’s multiple comparisons test. Means ± SD. **** indicates *p* < 0.001.

**Figure 3 biomedicines-10-01057-f003:**
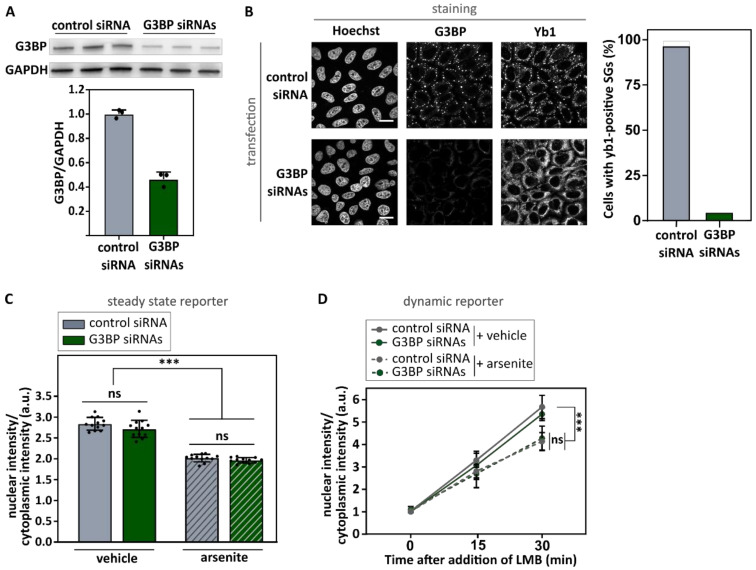
Inhibiting stress granule formation via knock-down of G3BP1 and G3BP2 does not rescue arsenite-induced import deficits. (**A**). HeLa Kyoto cells were transfected with both G3BP1- and G3BP2-specific siRNA or control siRNA. As a proof of concept, the G3BP1 level was determined using Western blot 48 h after transfection. Glyceraldehyde 3-phosphate dehydrogenase (GAPDH) was used as loading control and the relative G3BP1 level was determined. Data are shown as mean ± SD of three biological replicates. (**B**). Cells transfected with both G3BP1- and G3BP2-specific siRNA or control siRNA were treated with 500 µM arsenite for 60 min and subsequently stained with indicated antibodies. Hoechst was used to visualize the nucleus. Knock-down of G3BP1 and G3BP2 inhibited stress granule formation (indicated by the absence of G3BP1-positive and yb1-positive granules). Scale bar = 20 µm. (**C**,**D**). HeLa Kyoto cells expressing the NLS_cmyc_-GFP_2x_-NES_ikb2_ reporter (**C**) or the NLSSV40-mNeonGreen_2x_-NES_pki_ reporter (**D**) were transfected with both G3BP1- and G3BP2-specific siRNA or control siRNA. After 48 h, cells were treated with arsenite (or vehicle), before measuring the nuclear and cytoplasmic intensity of the reporter. Data were automatically analyzed (High Content Imager). No rescue of the arsenite-induced transport deficit was observed. Kruskal–Wallis test followed by Dunn’s multiple comparisons test. Data are shown as geometric mean ± geometric SD. N = 12 (**C**)/9 (**D**) (four (**C**)/three (**D**) experiments with each three wells, 1000 to 3000 cells per well). ns: no significant difference; *** indicates *p* < 0.005.

**Figure 4 biomedicines-10-01057-f004:**
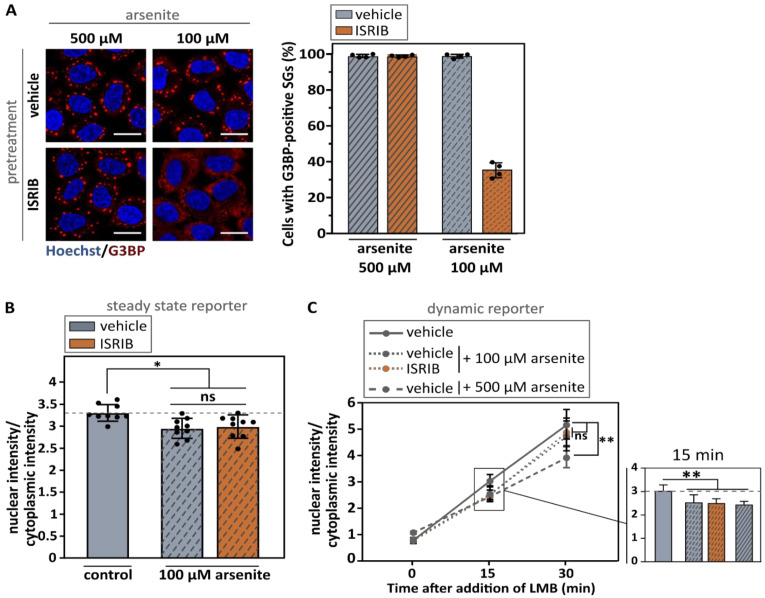
Suppressing stress granule formation via inhibition of the integrated stress response does not rescue arsenite-induced import deficits. (**A**). HeLa Kyoto cells were pre-incubated with 2 µM ISRIB (or vehicle) 4 h before treatment with the indicated arsenite concentrations. Cells were fixed with 4% PFA and stained with an anti-G3BP antibody. Stress granule formation was suppressed in cells treated with 100 µM arsenite, but not with 500 µM arsenite. The percentage of cells containing G3BP-positive granules was counted in four experiments. Each dot represents the average of one experiment with 50–200 cells per experiment. Mean ± SD. Scale bar = 20 µm. (**B**,**C**). HeLa Kyoto cells expressing the NLS_c-myc_-GFP_2x_-NES_ikb2_ reporter (**B**) or the NLS_SV40_-mNeonGreen_2x_-NES_pki_ reporter (**C**) were incubated with 2 µM ISRIB (or vehicle). After 4 h, cells were treated with indicated concentrations of arsenite, before measuring the nuclear and cytoplasmic intensity of the reporter. A zoom in on time point 15 is shown in the bar graph of panel (**C**). Data were automatically analyzed (High Content Imager). No rescue of arsenite-induced transport deficits was observed. Kruskal–Wallis test followed by Dunn’s multiple comparisons test. Data are shown as geometric means ± geometric SD. N = 6–9 (three experiments with each three wells, 1000 to 2000 cells per well—two experiments for 500 µM arsenite). ns: no significant difference; * indicates *p* < 0.05; ** indicates *p* < 0.01.

**Figure 5 biomedicines-10-01057-f005:**
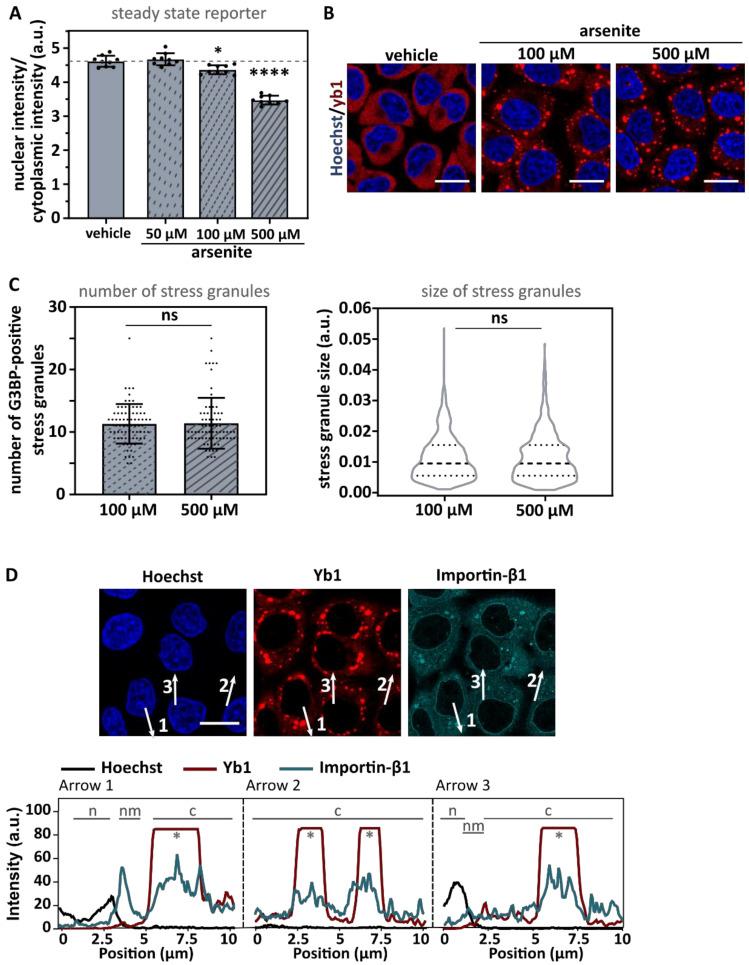
Dose-dependent arsenite nucleocytoplasmic transport deficits do not correlate with size and number of stress granules. (**A**). HeLa Kyoto cells expressing the reporter NLS_c-myc_-GFP_2x_-NES_ikb2_ were treated with the indicated concentrations of arsenite before fixation. Nuclear and cytoplasmic intensity was automatically measured in three replicates (three wells per experiment with 100–2000 cells per well). Both 100 µM and 500 µM or arsenite resulted in a decreased nuclear intensity, with a stronger decrease in cells treated with 500 µM. Mann–Whitney U test. Geometric means ± geometric SD. * indicates *p* < 0.05; **** indicates *p* < 0.001. (**B**). HeLa Kyoto cells treated with the indicated concentrations of arsenite and stained with anti-yb1 antibody. Scale bar = 20 µm. (**C**). HeLa Kyoto cells were treated with the indicated concentrations of arsenite, fixed and stained with an anti-G3BP antibody. The number of G3BP-positive stress granules per cell (left graph) and size of G3BP-positive stress granules (right graph) were measured in ImageJ. No difference in the number of stress granules per cell or size of stress granules was observed. Left graph: each dot represents one cell, with n = 75 from three experiments. Unpaired *t*-test. Mean ± SD. Right graph: truncated violin plot of individual stress granules with approximately 800 stress granules from three experiments. Mann–Whitney U test. ns: no significant difference. (**D**). Representative images of HeLa Kyoto cells treated with 100 µm arsenite and stained with an anti-G3BP and anti-importin-β1 antibody. Intensity plots are shown and confirm the localization of importin-β1 at yb1-positive stress granules. n = nucleus; nm = nuclear membrane; c = cytoplasm; * = stress granule. Scale bar = 20 µm.

**Figure 6 biomedicines-10-01057-f006:**
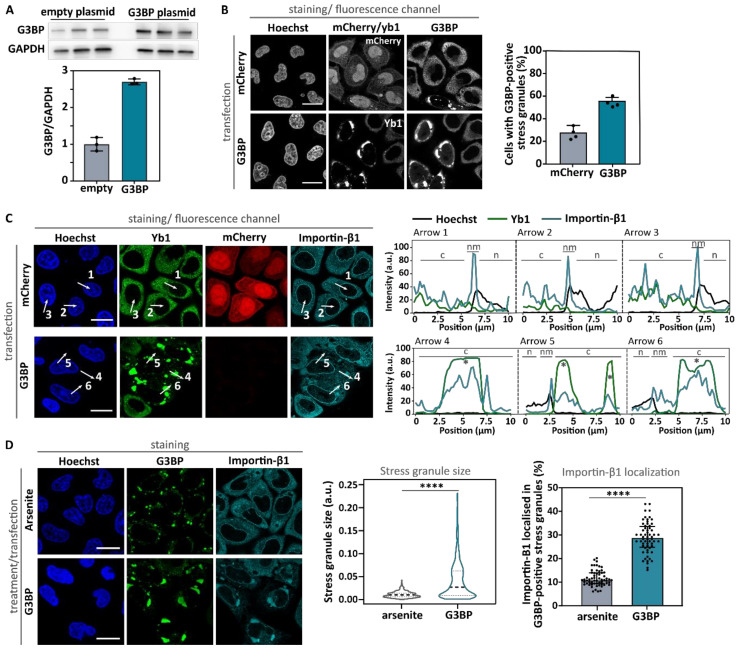
G3BP overexpression results in large stress granules and a strong importin-β1 co-localization. (**A**). HeLa Kyoto cells were transfected with plasmids expressing mCherry or G3BP1. At 48 h after transfection, G3BP overexpression was determined by Western blot. Relative expressions of G3BP compared to the loading control GAPDH are shown as mean ± SD of three biological replicates. (**B**). Cells transfected with the mCherry or G3BP plasmids were fixed at 48 h after transfection and stained with indicated antibodies. The percentage of cells with stress granules is presented. Each dot is the average of one experiment with 193–292 cells per experiment. Means ± SD. (**C**). Intensity plots confirming colocalization between yb1 and importin-β1. n = nucleus; nm = nuclear membrane; c = cytoplasm; * = stress granule. Scale bar = 20 µm. (**D**). Comparison of stress granule size and importin-β1 localization induced by arsenite treatment or G3BP overexpression. Left graph: Size of stress granules was measured using ImageJ in 75 cells of three experiments (25 cells per experiment). Stress granules were significantly larger in G3BP overexpression cells. Truncated violin plot. Mann–Whitney U test. Right graph: Percentage of importin-β1 intensity in stress granules compared to the whole cell intensity was manually measured in arsenite-treated cells and G3BP overexpression cells. Significantly more importin-β1 localized to stress granules in G3BP overexpression cells. Each dot represents one cell with n = 60 from three experiments (20 cells per experiment)**.** Mann–Whitney U test. Median ± IQR. Scale bar = 20 µm. **** indicates *p* < 0.001.

**Figure 7 biomedicines-10-01057-f007:**
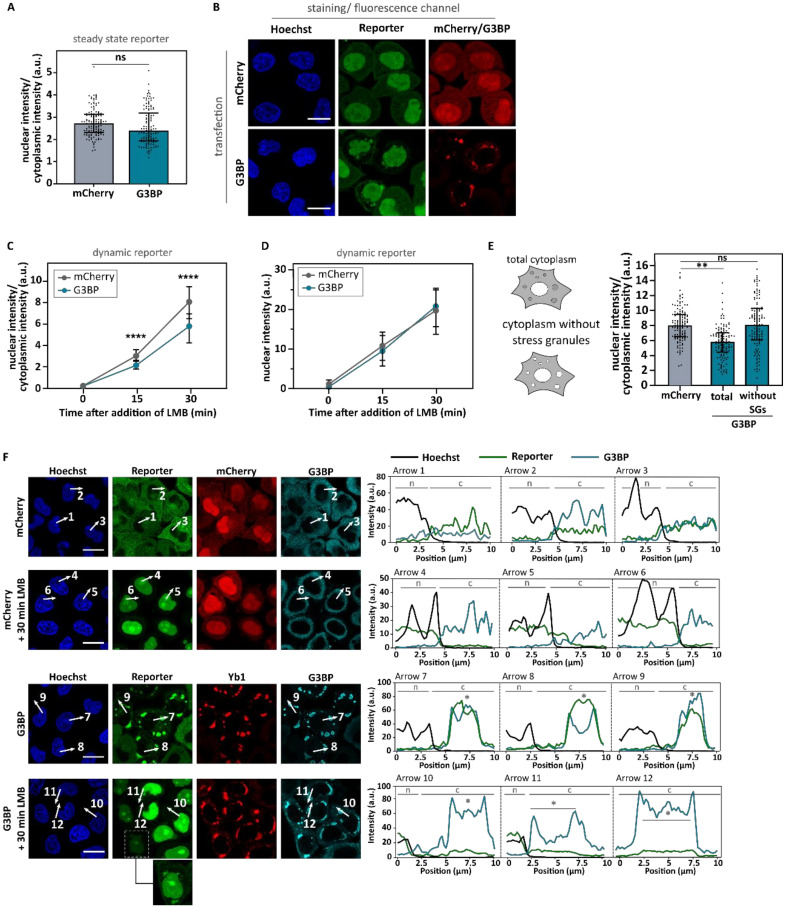
Induction of stress granules via the overexpression of G3BP1 does not impede nuclear import. **(A**). HeLa Kyoto cells expressing the NLS_c-myc_-GFP_2x_-NES_ikb2_ reporter were transfected with mCherry or G3BP1-expressing plasmids. After 48 h, nuclear and cytoplasmic intensity of the reporter was measured in mCherry-expressing cells without stress granules and G3BP-expressing cells with stress granules. No significant decrease in nuclear import was observed. Confocal images were analyzed using ImageJ. Data of three experiments were used. N = 90 of which each data point is one cell (30 cells per experiment). Mann–Whitney U test. Medians ± IQR. Ns: no significant difference. (**B**). Representative images of NLS_c-myc_-GFP_2x_-NES_ikb2_ cells used in panel (**B**). Despite the strong localization of the reporter at G3BP overexpression-induced stress granules, no decrease in nuclear intensity is observed. Scale bar = 20 µm. (**C**). HeLa Kyoto cells expressing the NLS_SV40_-mNeonGreen_2x_-NES_pki_ reporter were transfected with mCherry- or G3BP1-expressing plasmids. After 48 h, nuclear import was measured in mCherry-expressing cells without stress granules and G3BP-expressing cells with stress granules. A significant decrease in nuclear/cytoplasmic ratio was observed. Confocal images were analyzed using ImageJ. Data of four experiments were used. N = 120 of which each data point is one cell (30 cells per experiment). Mann–Whitney U test. Medians ± IQR. **** indicates *p* < 0.001. (**D).** Nuclear intensity of cells used in panel (**B**). No significant decrease in nuclear intensity was observed. Mann–Whitney U test. Medians ± IQR. (**E**). Data obtained in panel (**C**) after 30 min treatment with LMB were reanalyzed by excluding the stress granules when measuring the cytoplasmic intensity. When the stress granules were excluded, no nucleocytoplasmic transport deficits were observed. Kruskal–Wallis test followed by Dunn’s multiple comparisons test. Medians ± IQR. Ns: no significant difference; ** indicates *p* < 0.01. (**F**). Representative images and corresponding intensity plots of NLS_SV40_-mNeonGreen_2x_-NES_pki_ reporter cells transfected with mCherry or G3BP and treated with the indicated conditions. Intensity plots indicate that the reporter colocalizes with G3BP at G3BP overexpression-induced stress granules (arrow 7–12). n = nucleus; nm = nuclear membrane; c = cytoplasm; * = stress granule. Scale bar = 20 µm.

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
