# Peer review of "Cellular Stress Induces Nucleocytoplasmic Transport Deficits Independent of Stress Granules"

_biomedicines, 2022, doi:10.3390/biomedicines10051057_

Round 1

Reviewer 1 Report

The study is generally well written and perfectly designed. The Authors investigated whether there is a causal link between stress granule formation and nucleocytoplasmic transport deficits. The obtained results are interesting and may have an important impact on the development of therapeutic strategies. The applied methods are adequate and the results are clearly presented.

Minor comments:

In my opinion, the Introduction should be more concise. The manuscript requires editorial proofreading.

Reviewer 2 Report

Manuscript ID: biomedicines-1684023

Cellular stress induces nucleocytoplasmic transport deficits independent of stress granules

Under various stress conditions, cells inhibit translation as one of the survival responses, and subsequently, stress granules are formed in the cytosol. At the same time, nucleocytoplasmic transport is inhibited by cellular stress. Since the proteins involved in the nucleocytoplasmic transport machineries are found in the stress granules, the formation of stress granules has been considered to play a causative role for the disruption of the nucleocytoplasmic transport in several neurodegenerative disorders. However, the authors report in the present manuscript that the stress granules do not cause nucleocytoplasmic transport deficits directly. To access the nucleocytoplasmic transport, the authors used cell lines stably expressing fluorescent reporters, and uncoupled cellular stress from stress granule formation using pharmacological inhibitor or by modulating the expression level of G3BP1/2.

This manuscript is well-organized, although the authors could not address the alternative cause for the inhibition of nucleocytoplasmic transport in stressed cells. The following points need to be clarified to avoid misunderstanding.

  1. In some figures (e.g., Fig. 2B, C, D, Fig. 3C, D, Fig. 4B, C, and so on), the fluorescence intensity of the reporters is shown only in quantified graph. Please include some typical fluorescence images of reporter cells.

  1. In most of the figures (e.g., Fig. 1A, B, Fig. 2E, Fig. 5D, and so on), co-localization of signals is shown only by the graph of fluorescence intensity. Please add merged figures of the signals of interest.

  1. In Fig. 3A, including the legend, the terms G3BP, G3BP1, and G3BP2 are confusing. In Fig. 3A, western blotting of G3BP is shown. Because both B3BP1 and G3BP2 are knocked-down, the results of these two proteins need to be shown.

  1. In the discussion, line 560-561, it is hard to follow the conclusion, because cells expressing poly-PR100 -SGs show no difference from control cells (Fig. S7).

Reviewer 3 Report

Vanneste and colleagues present data supporting the independence of stress induced nucleocytoplasmic trafficking defects from stress granule formation. Using some elegant in vitro cell culture assays, the authors tested the hypothesis that there exist a causal link between stress granule formation and nucleocytoplasmic trafficking defects in neurodegenerative diseases, such as ALS. By genetically and/or pharmacologically uncoupling stress granule formation from stress, the authors showed that nucleocytoplasmic trafficking defects still occur, suggesting that unlike previous suggestions, trapping of nucleocytoplasmic transport proteins in stress granules is NOT required for trafficking defects across the nuclear membrane.

This is well written and well conceptualized manuscript with carefully performed experiments. The data presented support the authors conclusions and only minor comments/suggestions are noted below.

Introduction.

The introduction is a bit lengthy but at the same time misses sometimes detailed information to better inform the reader on the scientific background/rationale for the proposed hypothesis. E.g. line 44/45 – this statement could have been expanded on in more detail given that the authors hypothesis is based on this information.

Methods.

Line 133-135 – how do the authors justify using the average of one microscopic image as one data point? Would one not consider using one cell as one data point?

Line 136 – have the authors demonstrated that their high content imager generates equal data sets than their manual data analyses? If yes, this should probably be cited here.

Line 201 – the authors should describe how they quantify their western blot data. Further, an uncropped western blot should be provided in the supplement.

Results.

Line 222 - The reviewer is wondering if the authors actually know which specific importin alpha is responsible for the import of their reporter construct? This information could be helpful in fully understanding whether their assay only measures import through importin alpha 1 or whether other importins are involved in the measured import as presented.

Line 224 – did the authors actually look at TDP-43 co-localization to stress granules?

Line 232 – figure 2B – it would be great to support this quantification with some images, maybe in the supplement (same for Figures 2 C+D).

Line 284 – this is in reference to a question from above – are the authors certain that only importin alpha 1 is involved in this transport – if not, could stress potentially switch the use of importin alpha subtypes?

Line 295 – While the NLS-lacking GFP reporter shows reduced localization to stress granules, it still does co-localize with G3BP. Can the authors speculate on why this might be the case?

Line 305 – this statement should be revisited and maybe toned down given that there is still reporter protein present in the nucleus suggesting that some nuclear import might still be occurring. The authors should acknowledge this and if needed discuss their conclusions in more detail.

Line 317 and 353 – here, the authors should refer to supplemental figure 6 – without that figure, the reviewer immediately wondered if it would be helpful to stain for importins under the G3BP knockdown conditions, to determine if there is stress-induced aberrant nuclear pore protein machinery expression even in the absence of stress granules – which is exactly what the authors did in S6.

Line 396 – what is known about the stress granules that are formed due to overexpression of G3BP1? Are they similar to ‘regular’ stress granules?

Line 401 – what happens to importin alphas?

Line 427 – the authors should reference figure 6D here

Minor overall comments.

The font seems to change at varying locations throughout the manuscript.

Supplemental figure 6 – image panels should be labeled with importin alpha 1 (not 2), correct?
